# The impact of size on middle-ear sound transmission in elephants, the largest terrestrial mammal

**Caitlin E. O'Connell-Rodwell**[1,2]*, **Jodie L. Berezin**[1], **Anbuselvan Dharmarajan**[1], **Michael E. Ravicz**[1,2], **Yihan Hu**[1], **Xiying Guan**[3], **Kevin N. O'Connor**[1], **Sunil Puria**[1,2,4]

1 Eaton-Peabody Laboratories, Massachusetts Eye and Ear, Harvard Medical School, Boston, Massachusetts, United States of America, 2 Department of Otolaryngology, Head & Neck Surgery, Harvard Medical School, Boston, Massachusetts, United States of America, 3 School of Medicine, Wayne State University, Detroit, Michigan, United States of America, 4 Graduate Program in Speech and Hearing and Biosciences and Technologies, Harvard Medical School, Boston, Massachusetts, United States of America

* Caitlin_OConnell@meei.harvard.edu

**Data Availability Statement:** All relevant data are within the manuscript and its Supporting information files.

## Abstract

Elephants have a unique auditory system that is larger than any other terrestrial mammal. To quantify the impact of larger middle ear (ME) structures, we measured 3D ossicular motion and ME sound transmission in cadaveric temporal bones from both African and Asian elephants in response to air-conducted (AC) tonal pressure stimuli presented in the ear canal ($P_{EC}$). Results were compared to similar measurements in humans. Velocities of the umbo ($V_U$) and stapes ($V_{ST}$) were measured using a 3D laser Doppler vibrometer in the 7–13,000 Hz frequency range, stapes velocity serving as a measure of energy entering the cochlea—a proxy for hearing sensitivity. Below the elephant ME resonance frequency of about 300 Hz, the magnitude of $V_U/P_{EC}$ was an order of magnitude greater than in human, and the magnitude of $V_{ST}/P_{EC}$ was 5x greater. Phase of $V_{ST}/P_{EC}$ above ME resonance indicated that the group delay in elephant was approximately double that of human, which may be related to the unexpectedly high magnitudes at high frequencies. A boost in sound transmission across the incus long process and stapes near 9 kHz was also observed. We discuss factors that contribute to differences in sound transmission between these two large mammals.

## Introduction

Extensive comparative studies across a wide range of classes and species have shown that terrestrial mammalian middle ears have similar structures that scale with skull size [1–3], spanning from the shrew (smallest) to the elephant (largest). The mammalian middle ear (ME) influences hearing ability, as it serves as an impedance-matching device from the low-density air-filled ear canal to the high-density fluid-filled cochlea [4–6] to aid in sound transmission.

The velocity of the umbo ($V_U$) is a measure of input into the ossicular chain, the umbo being the most depressed part of the inward projection of the tympanic membrane (TM), or

**Funding:** Work supported by K01 DC017812 from the NIDCD of the National Institute of Health (to CEO-R) and The Amelia Peabody Fund (to SP). The funders had no role in study design, data collection and analysis, decision to publish, or preparation of the manuscript. https://grants.nih.gov/ https://ameliapeabody.org/.

**Competing interests:** The authors have declared that no competing interests exist.

eardrum, which is attached to the arm of the malleus, and the stapes velocity ($V_{ST}$) is a measure of input into the cochlea. $V_{ST}/V_U$, therefore, is the "velocity transfer ratio," a measure of sound transmission through the ossicular chain. To understand how differences in size and ME sound transmission are related, particularly at low frequencies, we chose to study the magnitude of ossicular velocities in the largest terrestrial middle ear, the elephant.

For context, the elephant eardrum is at least 7x larger than the human eardrum in area, and elephant ossicles have approximately 10x more mass than human ossicles [1, 2]. The mid-to-high frequency region of elephant hearing generally overlaps with the low-to-mid frequency region of human hearing. In addition, elephants are more sensitive at frequencies below about 100 Hz, while humans are more sensitive at frequencies above about 1 kHz [7].

Here, we present the first data on elephant ME ossicular motion and compare it to human. We measured the 3D motion of the malleus, incus, and stapes in postmortem elephant and human temporal-bone specimens for sound at frequencies from ~10 Hz to ~12 kHz. We quantify how the more-massive eardrum and ossicles of the elephant move in response to acoustic stimuli at very low frequencies, as well as at high frequencies, to understand how ossicular motion may be impacted by eardrum and ossicular-chain morphology, and the influence that morphology has on the input to the cochlea.

## Methods

### Postmortem materials

This study was performed in line with the principles of the Declaration of Helsinki. Human post-mortem specimens: Human-subject research in this manuscript meets the criteria for exemption from the Mass General Brigham IRB requirements in DHHS regulations (45 CFR 46), which would include participant consent. Specifically, the research falls under Exemption 4: Research involving the collection or study of existing data, documents, records, pathological specimens, or diagnostic specimens, if these sources are publicly available or if the information is recorded by the investigator in such a manner that subjects cannot be identified, directly or through identifiers linked to the subjects. Human temporal bones were harvested from the Massachusetts General Hospital morgue in accordance with the appropriate Mass Eye & Ear Otopathology Laboratory protocols. Elephant post-mortem specimens: No approval of research ethics committees was required to accomplish the goals of this study, because experimental work was conducted on post-mortem temporal bones donated from zoos and sanctuaries.

Measurements were made in the ears of four elephant temporal bones. One specimen was from an African elephant (*Loxodonta africana*) and three were from Asian elephants (*Elephas maximus*). The temporal bones were from different animals, labeled ETB2 for the African elephant (right ear, four-month-old female calf), and ETB3 (left ear, died one day before birth in utero male calf), ETB6 (right ear 72-year-old adult female), and ETB7 (left ear, 27-day-old male calf) for the Asian elephants. Each specimen was visually screened for any pathologies prior to experimentation. Although measurements from the single adult specimen (ETB6) were not available for the incus ($V_I/P_{EC}$) or stapes ($V_{ST}/P_{EC}$), the available ETB6 $V_U/P_{EC}$ measurements appear comparable to the three calf specimens.

Four human temporal bones were harvested from the Massachusetts General Hospital morgue in accordance with the appropriate Mass Eye & Ear Otopathology Lab protocols (Human Subjects Protocol # 2021P002348). All human specimens were obtained from donors with no history of otologic disease and were visually screened for ME pathologies on arrival and labelled as TB18 (left ear, gender and age not available), TB19 (left ear, 46-year-old male, collected on 04/02/2019), TB20 (left ear, 58-year-old female, collected on 06/22/2017), and TB24

(right ear, 41-year-old female, collected on 07/23/2019). The procurement date was not available for TB 18. We assume it was received from MGH shortly before we started dissections and experiments, which started on March 18, 2019.

All specimens were stored in the refrigerator in saline with a few drops of iodine as an antibacterial after preparation and between measurement sessions. After the completion of experiments, the specimens were triple-wrapped in 0.9% saline-soaked gauze, plastic food wrap, and aluminum foil, and placed in a -20˚ C freezer. For human, TB24 was used for imaging only; vibrometry was performed on TB18, TB19, and TB20. These stored frozen specimens were retrieved months or years later for anatomical or other measurements as needed.

## Specimen preparation

Both elephant and human temporal bones were dissected to fit within a bowl-shaped temporal-bone-specimen holder (for elephants the average was 280 cm$^3$), to which it could be affixed using adjustable screws for ease of manipulation during dissection and preparation. The ME cavity (MEC) in the human temporal bones was accessed through the posterior–inferior facial recess after opening the recess with an otologic drill. This allowed optical and mechanical access to the ossicles and the medial side of the TM, for placement of targets and to accommodate the three beams of the 3D laser Doppler vibrometer (3D LDV; CLV 300, Polytec, Germany). Similarly, the elephant MEC was accessed through the posterior–inferior approach by dissecting bone and tissue. From this approach, the view of the ossicles was blocked by a thick membrane-like tissue that divides the MEC. This tissue was dissected out to expose the ossicles. Further details on specimen preparation can be found in S1 Text.

The ear canal lengths differed relative to the age, sex and species of elephant and the ability to trim down the specimen in each case. The remaining ear canal length (to the lateral edge of the eardrum) was approximately 20–40 mm while for the human specimens, the remaining ear canal was about 10 mm long. One end of a 25–30-mm long threaded rod (size #10–32) was glued into the inferior bony portion of each temporal bone with dental cement and epoxy, away from the region of interest, and the other end was screwed into a mini-shaker (4810, Brüel+Kjær, Naerum, Denmark; Fig 1A) to accommodate a series of bone-conduction (BC) measurements not reported here.

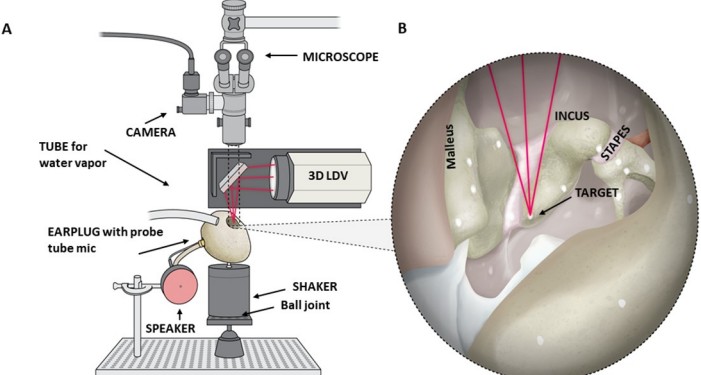

**Fig 1. Specimen preparation and experimental set up.** (A) Experimental setup of a speaker to generate ear canal sounds, probe-tube microphone, 3D LDV mounted on a motorized stage with a dichroic mirror, and temporal bone on a shaker mounted on a vibration isolation table. The microscope allows visualization of the specimen below through the dichroic mirror. (B) Cutaway of an elephant temporal bone with the three LDV laser beams aligned to a single point on a reflective bead placed on the incus.

## Experimental setup

The mini-shaker was mounted on a rotation–tilt ball joint that was attached to a breadboard mounted to a vibration isolation table (Nexus, T46H, Thorlabs, Newton, NJ), to minimize noise intrusion. This setup enabled the orientation of the temporal bone to be adjusted for optimal visualization and 3D LDV access of the ossicles through the opening into the MEC (Fig 1).

## Sound stimulation and ear-canal sound pressure

Sound stimulation was generated by a loudspeaker and delivered to the ear canal through a custom assembly (Fig 1A). For more details about the sound stimuli and microphone assembly, refer to S2 Text. The distance between the microphone probe-tube tip and the umbo ranged from 19–53 mm in the elephant specimens (ETB2 = 41; ETB3 = 53; ETB6 = 19; ETB7 = 45) and was about 5 mm in the human specimens.

## Measurements of 3D ossicular velocities

Four reflective tape targets were placed along each ossicle (see S2 Text; Figs 1B and 2A). The velocity of each target was measured in three orthogonal directions using a 3D LDV. The measured velocity components of the reflective targets on each ossicle were initially in terms of the reference frame of the 3D LDV: $V_X$ (left–right) and $V_Y$ (front–back) in the horizontal plane, and $V_Z$ (up–down) in an orthogonal vertical plane. The head of the LDV was mounted on an X, Y, Z motorized stage controlled by SyncAv. The three LDV beams were reflected by an angled dichroic mirror such that the effective Z axis was vertical (Fig 1). The 3D ossicular-motion measurement methods have been described previously [8] and in S2 Text and S1 Fig.

Final sample numbers for data presented for each specimen are as follows: ETB2, umbo (n = 1), I1 (n = 1), stapes (n = 2); ETB3, umbo (n = 2), I1 (n = 1), stapes (n = 2); ETB6, umbo (n = 2); ETB7, umbo (n = 3), I1 (n = 2), stapes (n = 5); TB18, stapes (n = 2); TB19, umbo

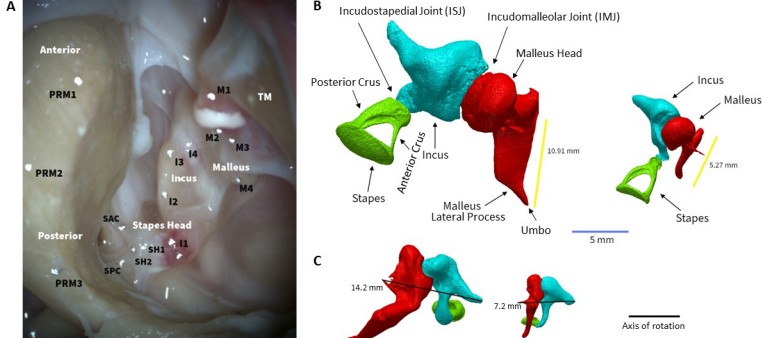

**Fig 2. Elephant ossicles in the middle ear cavity, and μCT images of elephant and human ossicles.** Elephant temporal bones were prepared in order to open the middle ear cavity widely enough to access the ossicles in such a way that the three beams of the 3D LDV laser could hit the targets with no obstructions. (A) Image through the microscope after placement of targets. Reflective targets were placed on all three bones and the promontory: Four along the malleus from inferior at the umbo (M1), across the lateral process to superior, on the head of the malleus near the incudomalleolar joint (IMJ) (M4). Four along the incus long-process from superior (I4) near the IMJ to at the distal tip of the incus, close to the incudostapedial joint (ISJ) (I1). Two on the stapes head SH1,2 and three on the promontory (PRM) 1, 2, 3. (B) The malleus, incus and stapes reconstructed from μCT image for both the elephant (left) and human (right). Estimated malleus lengths are shown in yellow. (Image from ETB1 and μCT from ETB2 and TB24; refer to Table 1 for measurements). (C) Additional view of the axis of rotation (solid black line).

(n = 1), I1 (n = 1), stapes (n = 2); TB20, umbo (n = 1), I1 (n = 1), stapes (n = 2). Overall total measurements per ossicle per group: elephant umbo = 8, I1 = 4, stapes = 9; human umbo = 2, I1 = 2, stapes = 6). See S1 Data for raw velocity measurements.

## Projection of 3D velocity components onto anatomically relevant directions

The anatomically relevant directions of target velocities (for the stapes and incus, the piston direction perpendicular to the footplate; for the umbo, the direction perpendicular to the tympanic ring) were not aligned with the native axes of the 3D LDV (see S3 Text for a detailed description). To project the measured motion components onto the relevant anatomical directions, coordinate transformations between the LDV reference frame and the desired reference frames were computed using rotation matrices calculated based on measured angles between the desired specimen axes and LDV axes (S1 Table and S2 Fig). From these rotation matrices, the 3D velocities were projected onto the desired 1D anatomical directions (see S4 Text for a detailed description). This approach allowed us to determine the velocities in directions that were not possible to access and measure directly. Reported measurements are 1D that have been projected onto their anatomically relevant directions (see S1 Fig and S2 Data for processed velocity measurements).

## Micro-CT imaging and segmentation

To compare anatomical differences, some temporal bones were imaged using a Micro-CT X-ray microscope scanner (Zeiss, Xradia 520 Versa), from which segmented 3D-volume reconstructions were generated of the TM and ossicles using Synopsis Simpleware Software (Mountain View, CA). One specimen each of African elephant (ETB2), Asian elephant (ETB6), and human (TB24) were scanned. The scan resolution for the elephant ears was about 25.2 μm and for human it was 15.1 mm. Segmentation of the TM was performed, and stacks of segmented slices were then combined to construct the 3D volume of the TM. For the ossicles, automatic thresholding techniques were used to segment and reconstruct each bone (e.g., Fig 2B and S1B Fig).

Middle ear morphometry was measured in Simpleware. TM surface area was calculated using the same methods as Nummela [2]. Using the "measurements" tool, two diameters were measured, the longest diameter and one perpendicular. The TM area was obtained by multiplying the radius from the two diameters and π, and the TM volume obtained by multiplying the volume by the thickness. All masses were calculated by using the formula, mass = volume*density. For the TM, density of water (1 mg/mm$^3$) was assumed. Malleus, incus, and stapes masses were calculated using the density value for the human. Density values from Sim & Puria [9] for human malleus (2.31 mg/mm$^3$), and incus and stapes (2.14 mg/mm$^3$) were used. Nummela [2] reported human, seal, and elephant ossicle densities between 2.0 mg/mm$^3$ and 2.3 mg/mm$^3$, which is similar to those from Sim & Puria [9] for human. Since the specific densities were not listed per species, we assumed the densities of elephants and humans were similar enough and used the values presented in Sim & Puria [9] for elephant calculations.

Malleus and incus lengths were calculated using recent methodologies [9]. A line was drawn through the axis of rotation, from the thin process of the malleus to the long arm of the incus (Fig 2C). Lines were then drawn from the tips of the ossicles towards the approximate center-right of the ossicles, meeting the axis of rotation. Lenths were also calculated using the Hemilä et al. [1] methodology to confirm consistency of measurements.

## Lever ratio and ossicular-velocity transfer functions

Functionally, a lever increases force while reducing displacement (or velocity; [4, 6]). Forces are not easy to measure *in-situ*, so we use velocity to describe the umbo-to-stapes-head velocity transfer function ($TF_{US} = V_{ST}/ V_U$), which is related to the reciprocal of the anatomical lever ratio (e.g., if the anatomical lever ratio were 3, defined as the ratio of the lengths of the malleus and incus lever arms, $l_M/l_I = 3$, then $TF_{US}$ would be expected to yield a value of 0.33). From our stapes and umbo velocity measurements, we calculated $TF_{US}$ for both human and elephant. We also measured $V_I$ in a subset of animals, which allowed us to decompose $TF_{US}$ into the product of two components: $TF_{US} = TF_{UI}*TF_{IS}$, such that $TF_{UI} = V_I/V_U$, a measure of malleus-incus complex (MIC) transmission that includes IMJ slippage; and $TF_{IS} = V_{ST}/V_I$, a measure of incus to stapes transmission that includes ISJ slippage.

## Umbo- and stapes-velocity group delays

One way to interpret phase accumulation in the $V_U/P_{EC}$ and $V_{ST}/P_{EC}$ transfer functions is by calculating the rate of change of the phase, which (multiplied by -1) is a measure of the group delay (GD). The GDs from the ear-canal microphone to different points on the ossicular chain can be used to examine the effects of flexibility of ME substructures, which include bending of the ossicles and slippage across the IMJ and ISJ [10]. Stiffer structures contribute smaller delays than structures with greater flexibility.

We calculated GD by obtaining a least-squares straight-line fit to the velocity/pressure phase (in cycles) in individual ears, when plotted on a linear frequency axis. In frequency ranges where the slope is constant, the negative of the slope of this fitted line with respect to frequency is the GD. The GDs were calculated for two frequency regions: (1) between 2 octaves above the ME resonance and 11 kHz "High frequency (HF)"; and (2) over the combined regions of HF and near resonance (NR) "Wideband" (WB). For elephant, the NR range was 0.6–1.2 kHz and the HF range was 1.2–11 kHz; for human, the NR range was 2–4 kHz and the HF range was 4–11 kHz. Since the $V_U/P_{EC}$ and $V_{ST}/P_{EC}$ phase includes a contribution from the acoustic propagation time between the probe tube tip and the umbo, this propagation time, equal to the speed of sound (344 m/s) divided by the distance between probe tube tip and umbo (19–53 mm, depending on ear), was subtracted from the estimated GD. The corrected GDs from individual ears and their mean and standard deviation (SD) are reported for $V_U/P_{EC}$ and $V_{ST}/P_{EC}$ in each group.

## Statistical analysis

We assessed whether $V_U/P_{EC}$ and $V_{ST}/P_{EC}$ magnitude and phase differed significantly across frequencies, between groups (elephants and humans) and ossicles (umbo and stapes) using linear mixed-effects models. For each specimen, we used the mean umbo and stapes velocity measurements (compiled in S2 Data). Magnitude data were log-transformed, and due to negative values (minimum value -3.49), phase data were transformed by taking the log of the phase plus four.

A total of eight linear mixed-effects models were performed using the 'lmer' function in the R package lmerTest [11]. First, we assessed the overall differences between human and elephant velocity, across the umbo and stapes using two models, one for the magnitude and the other for phase. Both models included a fixed-effects interaction between group and ossicle. To account for repeated measures, a nested, random effect of frequency within each specimen was included. Next, we assessed the umbo and stapes velocities in four separate models (one per ossicle, magnitude and phase), using frequency categories: low, between 17–1000 Hz; and high, greater than 1000 to 11000 Hz. For direct comparisons, all frequencies below 17 Hz (the

minimum human frequency collected) and above 11 kHz (the maximum frequency collected for humans) were removed. All four models included a fixed-effects interaction between group and frequency category and a random-effect of specimen to account for repeated measures. The final two models were used to assess differences in magnitude across the ossicle chain for elephants and humans, respectively. Both models included an interaction between frequency category and ossicle, with a random-effect term of specimen. Model assumptions of linearity, homoscedasticity, and normality were visually assessed; assumptions were met for all models (see S1 Code for details).

Statistical analyses were performed using RStudio [12] (version 2023.06.1) and R [13] (version 4.3.1). All models were assessed at α = 0.05.

## Results

### Anatomical comparisons

For context, we first summarize ME anatomical differences between elephant and human (Fig 2B and Table 1). The average TM surface area in the African elephant calf and Asian elephant adult (see S1B Fig) is about 7.5x larger than that of human (504.9 versus 67.3 mm$^2$), whereas it was reported as 6.6x larger (454 vs. 68.3 mm$^2$) based on measurements by Nummela [2]. These differences are likely due to varying ages between specimens. The African elephant TM surface areas was 1.5x larger than the Asian elephant (606.3 versus 403.6 mm$^2$), similar to Nummela's

**Table 1. Comparison of middle ear morphometry across elephant and human.**

| Dimensions | Our Measurements | | | | Previous Publications | | |
|---|---|---|---|---|---|---|---|
| | African Elephant | Asian Elephant | Human | Size Difference | African Elephant | Asian Elephant | Human |
| TM radius (mm) | 14.95 | 12.34 | 4.99 | 2.7x | - | - | - |
| TM surface area (mm$^2$) | 606.3 | 403.6 | 67.3 | 7.5x | 855 | 454 | 68.3 |
| TM thickness (μm)[a] | 424.2 | 361.2 | 222.6 | 1.8x | - | - | - |
| TM volume (mm$^3$)[a] | 211.4 | 154.0 | 12.7 | 14.4x | - | - | - |
| TM mass (mg)[a] | 211.4 | 154.0 | 12.7 | 14.4x | - | - | - |
| Malleus mass (mg) | 232.8 | 354.5 | 26.4 | 11.1x | 278 | 335.1 | 28.5 |
| Incus mass (mg) | 174 | 260.6 | 26.5 | 8.2x | 237 | 285.2 | 33.6 |
| Stapes mass (mg) | 13.1 | 23.4 | 3.6 | 5.1x | 22.6 | 27.2 | 2.5 |
| Ossicular chain mass (mg) | 419.9 | 638.5 | 56.5 | 9.4x | 537.6 | 647.5 | 64.6 |
| Malleus length (mm) | 14.44 | 14.44 | 4.69 | 3.1x | - | 16.3* | 6.24* |
| Incus length (mm) | 4.78 | 5.54 | 3.27 | 1.6x | - | 8.50* | 4.46* |
| Lever arm (mm) | 19.22 | 19.98 | 7.96 | 2.5x | - | 24.8* | 10.7* |
| Lever ratio | 3.0 | 2.6 | 1.4 | 2.0x | - | 1.9 | 1.4 |

Our data is presented in the left three columns: ETB2, the four-month-old African elephant; ETB6, the 72-year-old Asian elephant; and TB24 for human. ETB6 TM and stapes were not completely segmented, with approximately 10% missing. For comparison, African elephant, Asian elephant, and human data from Nummela [2] and Hemilä et al. [1] are presented in the right two columns (Hemilä et al. [1] denoted with asterisks). Size differences were calculated using the average of ETB2 and ETB6 measurements. Ossicular chain mass was calculated as the total mass for the malleus, incus, and stapes. For ease of comparison, we provide this value for the Nummela data. The difference between our calculation of the African elephant and previous publications may be due to the younger age of our specimen (the African elephant age was not reported for previous publications).

[a] Calculating the thickness of the TM using μCT is known to be not very accurate due to the low x-ray absorption of soft tissues resulting in poor contrast [14]. Since these elephant specimen measurements are rarely made, we choose report what we currently have. The thickness of the human TM has been reported to have a wide range (from 30–150 μm), with many researchers adopting an average human TM thickness of 74–100 μm (discussed in [15]). Therefore, the TM thickness, volume, and mass calculations reported in Table 1 are likely inflated by about 2–3 times than what would be obtained using OCT methods. However, the ratios of those values between elephant and human should be representative.

[2] findings (2x larger, 855 versus 454 mm$^2$), suggesting possible differences between species. Given that African elephants height and weight are about 15–20% more than Asian elephants on average, it's likely their ears would follow a similar trend. We report the thickness estimates in Table 1, where the elephant TM is about 1.8x thicker than the human TM (details in table caption). The calculated volume and mass of the elephant TM was 14.4x that of human.

The combined malleus and incus mass was nearly 10x heavier for elephant than human (511 vs. 53 mg), while the combined mass from Nummela [2] for elephant was 9x heavier. Averaging the two results, we obtain a malleus–incus mass that is 9.5x larger for elephant than for human. The stapes mass is estimated to be about 1/22$^{nd}$ of the malleus-incus masses for both groups.

Our estimates of ossicular mass (combined malleus, incus, and stapes) are quite similar (within 15%) to those obtained by Nummela [2] by different methods. Like Nummela, we found a higher ossicular mass in Asian elephant specimens (639 mg) than the African elephant (420 mg), though the African elephant specimen was a calf, while the others were mature adults. Our estimate in human specimens (57 mg) is quite similar to Nummela's (64 mg) as well. Both studies showed African elephant ossicular mass 7–8x that of human and Asian elephant 10-11x that of human.

The anatomical lever ratio was computed as the ratio of the malleus and incus lever arms, where the malleus lever arm is defined as the distance from the umbo to the presumed axis of malleus–incus rotation. The incus lever arm is defined as the distance from the same rotation axis to the bend in the incus just before it attaches to the stapes head, based on our μCT measurements in ETB2, ETB6 and TB24 (Table 1). The anatomical axis of rotation is usually estimated as passing through the anterior-malleolar ligament and the posterior-incudal ligament (Fig 2C). We found a lever ratio in elephants of 2.8, and humans having a ratio of 1.4, based on our μCT measurements in ETB2, ETB6 and TB24. The lever ratio for elephant from the present measurements is higher than that from Hemilä et al. [1] (Table 1). We calculated the $V_{ST}$/$V_U$ velocity transfer functions (TF$_{US}$), and at least at low frequencies their magnitudes are approximately equal to the reciprocal of the anatomic lever ratio of 2.8. Since our estimate of the lever ratio based on functional measurements (see next section) was also closer to about 3 in elephant, we chose to use 3 for the elephant lever ratio rather than 2 as reported in Hemilä et al. [1].

The cross-section of the malleus long process, which influences its stiffness, also differed between elephant and human. We looked at the cross-sectional area of the malleus long process in a plane orthogonal to a line between the umbo and the lateral process of the malleus. For elephant, this cross-section area is relatively thin and oblong, with much wider lateral–medial dimensions than for the human malleus whose cross-sectional area is rounder and more circular [16]. In addition, the neck of the malleus (from the lateral process to the head of the malleus) is relatively longer than that of the human. The stapes posterior crus of the elephant is wider and thicker than the anterior crus, whereas for human the two crura are somewhat more symmetric. In addition, the medial surface of the stapes footplate of the elephant is convex, whereas in the human it appears to be flat or concave (Fig 2B).

## Umbo and stapes velocities, normalized by ear-canal pressure

Fig 3 depicts a comparison of the mean and standard deviation (SD) of $V_U$/$P_{EC}$ and $V_{ST}$/$P_{EC}$ (magnitude in (mm/s)/Pa) in elephant (a, c) and human (b, d) over the measured frequency ranges. Because there is such a large variance between specimens, we also present the mean velocity measures per each temporal bone in Fig 4. Compared to elephant, human $V_U$/$P_{EC}$ and $V_{ST}$/$P_{EC}$ were lower in magnitude (p<0.001) and had less-negative phase (p<0.001). Further,

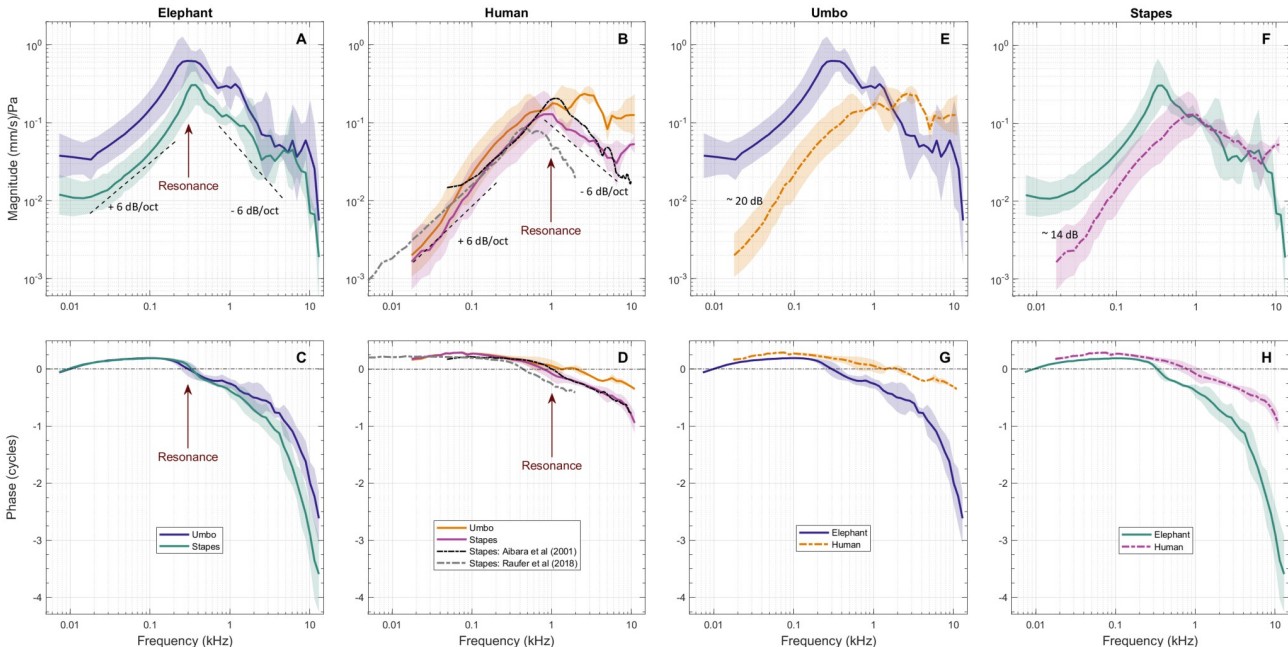

**Fig 3.** Mean and SD of both umbo and stapes velocities grouped by elephant (A) and human (B), and respective phases (C, D). Velocities are presented after projection on to the 1D piston directions and normalizing by the pressure measured near the eardrum (Ptm). The column with human data (B, D) includes previously published data by Aibara et al. [17] and Raufer et al. [18]. To facilitate ossicle motion comparison across elephant and human, data was grouped by umbo velocity (E, G) and stapes velocity (F, H). The top row (A, B, E, F) shows the magnitude in (mm/s)/Pa and bottom row (C, D, G, H) shows the unwrapped phase in cycles. Middle ear resonance is indicated where the phase crosses 0-cycles and there is a peak in the magnitude (~300 Hz for elephant and 1000 Hz for human). Shaded areas show +/- 1 SD. Number of temporal bones: elephant umbo (n = 4), elephant stapes (n = 3), human umbo (n = 2), human stapes (n = 3). Number of total measurements per ossicle per species (elephant umbo = 8, stapes = 9; human umbo = 2, stapes = 6).

there were significant differences in magnitude for both umbo (p<0.001) and stapes (p<0.001) below 1 kHz between the species. Above 1 kHz, the differences in magnitude were insignificant for umbo (p = 0.071) and stapes (p = 0.497). See Table 2 for the results of the statistical analyses.

In elephant, the mean $V_U/P_{EC}$ and $V_{ST}/P_{EC}$ magnitudes increased for frequencies above about 20 Hz with a slope of ~+6 dB/octave and reached a peak near approximately 300 Hz. Below 20 Hz, the magnitude tends to be nearly flat. The $V_U/P_{EC}$ magnitude was significantly greater than the $V_{ST}/P_{EC}$ magnitude (p = 0.036) by 6 dB in this frequency range (Fig 3A). Above the peak, the $V_U/P_{EC}$ and $V_{ST}/P_{EC}$ magnitudes decreased at about -6 dB/octave until about 3 kHz. Above 1 kHz the $V_U/P_{EC}$ was significantly higher than $V_{ST}/P_{EC}$ (p<0.001). Above 3 kHz, the $V_{ST}/P_{EC}$ magnitude flattened, and the $V_U/P_{EC}$ and $V_{ST}/P_{EC}$ magnitudes were similar. Below the magnitude peak at 300 Hz, the mean $V_U/P_{EC}$ and $V_{ST}/P_{EC}$ phases were approximately 0 cycles at the lowest frequencies and near +0.25 cycles at a few hundred Hz (Fig 3C). The phases of $V_U/P_{EC}$ and $V_{ST}/P_{EC}$ crossed 0 cycles at the frequency of the magnitude peak, consistent with an ME resonance at ~300 Hz.

The magnitude slope of ~+6 dB/octave and phase near +0.25 cycles in elephant are consistent with ME behavior controlled by stiffness in the 20 to 300 Hz frequency range, whereas the magnitude slope of -6 dB/octave and phase near -0.25 cycles between 400 and 600 Hz is consistent with ME behavior controlled by mass inertia above resonance. Below 20 Hz, the flat magnitude and phase less than ~0.1 cycles indicate resistive ME behavior. Above 1 kHz, the $V_{ST}/P_{EC}$ phase was more negative than the $V_U/P_{EC}$ phase, with the latter phase accumulating as

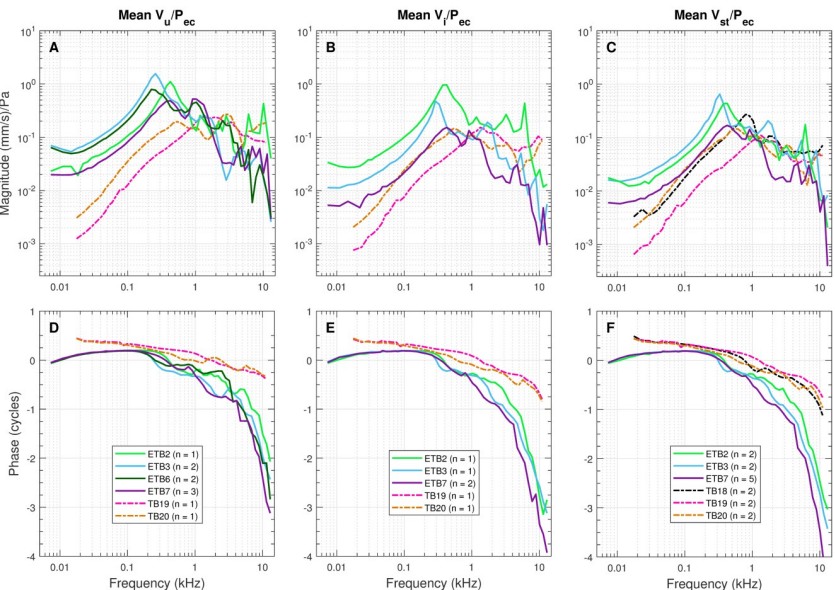

**Fig 4. Mean of piston direction velocities measured in individual ears of all elephant (solid lines) and human (dashed lines) temporal bone specimens.** (A, D) Depicts umbo velocity, $V_U/P_{EC}$. $V_I/P_{EC}$ measurements, depicted in the middle, were made at the end of the incus long process, after projecting onto the stapes piston direction (B, E). Stapes velocity, $V_{ST}/P_{EC}$ (C, F) is depicted in the third set of panels. Note that not all of the measurements were available for all of the temporal bones and thus the n's vary. We report these means to show that the range of each temporal bone between specimens was within the expected range, which, at least in human temporal bone, were about ±10 dB (e.g., Aibara et al. [17]). Additionally, these individual measurements were used to calculate ossicle transmission across joints ($TF_x$).

much as -2 cycles by 10 kHz while the former phase accumulated as much as -2.8 cycles over the same range. These large phase accumulations are consistent with ME delay (see next section). Note that for elephant measurements above about 5 kHz, the phase was unwrapped manually to ensure a continuous decrease in phase.

In human, the mean $V_U/P_{EC}$ and $V_{ST}/P_{EC}$ magnitude at low frequencies also increased with frequency (a mean low-frequency slope of ~+6 dB/octave), but its resonant peak occurred near 1 kHz instead. As in elephant, the $V_U/P_{EC}$ magnitude was greater than the $V_{ST}/P_{EC}$ magnitude (p = 0.050), but only by about a factor of ~1.6 in this frequency range (Fig 3B). Above resonance, the $V_U/P_{EC}$ magnitude was approximately flat with frequency, while the $V_{ST}/P_{EC}$ magnitude decreased by about -6 dB/octave (p<0.001). Below the resonance at 1 kHz, the mean $V_U/P_{EC}$ and $V_{ST}/P_{EC}$ phases were near +0.25 cycles (Fig 3D). As in elephant, the $V_{ST}/P_{EC}$ phase accumulated more than the $V_U/P_{EC}$ phase at higher frequencies, with the $V_U/P_{EC}$ phase reaching about -0.3 cycles and the $V_{ST}/P_{EC}$ phase reaching about -0.8 cycles by 10 kHz.

Fig 3B and 3D also show $V_{ST}/P_{EC}$ as measured in human ME temporal bones by Aibara et al. [17] from 50 Hz to 10 kHz, and by Raufer et al. [18] from 5 Hz to 2 kHz. There is a close correspondence between our $V_{ST}/P_{EC}$ measurements and those of Aibara et al. [17] from 100 Hz to 10 kHz, and those of Raufer et al. [18] between 50 and 500 Hz. These comparisons with previously published data test the applicability of 3D LDV methods and coordinate transformations by providing additional estimates of piston-direction velocities for comparison. The previous measurements were made by aligning a 1D LDV as close as possible to the stapes piston direction, usually within 30–45°, and then applying a 1/cos(θ) correction. The current $V_{ST}/P_{EC}$ measurements were determined by taking 3D velocity measurements of the stapes

**Table 2. Results of linear mixed-effect models.**

| Model | n | Predictors | Estimate | SE | *t* statistic | *p* |
|---|---|---|---|---|---|---|
| 1. Overall magnitude | 7(590) | Elephant, Stapes | -2.84 | 0.14 | | |
| | | Human | -0.73 | 0.18 | -3.99 | <0.001** |
| | | Umbo | 0.92 | 0.04 | 21.75 | <0.001** |
| | | Human*Umbo | -0.24 | 0.07 | -3.61 | <0.001** |
| 2. Overall phase | 7(590) | Elephant, Stapes | 1.25 | 0.02 | | |
| | | Human | 0.12 | 0.03 | 4.62 | <0.001** |
| | | Umbo | 0.07 | 0.01 | 7.65 | <0.001** |
| | | Human*Umbo | -0.04 | 0.01 | -2.71 | 0.007** |
| 3. Umbo magnitude | 6(296) | Elephant, High Freq. | -2.48 | 0.16 | | |
| | | Human | 0.58 | 0.28 | 2.05 | 0.071 |
| | | Low Freq. | 0.77 | 0.15 | 5.06 | <0.001** |
| | | Human*Low Freq. | -2.44 | 0.27 | -9.17 | <0.001** |
| 4. Umbo phase | 6(296) | Elephant, High Freq. | 1.15 | 0.01 | | |
| | | Human | 0.21 | 0.02 | 8.89 | <0.001** |
| | | Low Freq. | 0.26 | 0.01 | 18.1 | <0.001** |
| | | Human*Low Freq. | -0.16 | 0.02 | -6.49 | <0.001** |
| 5. Stapes magnitude | 6(294) | Elephant, High Freq. | -3.14 | 0.28 | | |
| | | Human | 0.29 | 0.40 | 0.73 | 0.497 |
| | | Low Freq. | 0.30 | 0.18 | 1.69 | 0.092 |
| | | Human*Low Freq. | -1.37 | 0.26 | -5.39 | <0.001** |
| 6. Stapes phase | 6(294) | Elephant, High Freq. | 0.97 | 0.03 | | |
| | | Human | 0.32 | 0.04 | 7.91 | <0.001** |
| | | Low Freq. | 0.43 | 0.03 | 14.96 | <0.001** |
| | | Human*Low Freq. | -0.27 | 0.04 | -6.55 | <0.001** |
| 7. Elephant magnitude | 4(350) | Stapes, High Freq. | -3.10 | 0.21 | | |
| | | Umbo | 0.62 | 0.18 | 3.47 | <0.001** |
| | | Low Freq. | 0.30 | 0.17 | 1.80 | 0.073 |
| | | Umbo*Low Freq. | 0.47 | 0.22 | 2.10 | 0.036** |
| 8. Human magnitude | 3(240)[a] | Stapes, High Freq. | -2.85 | 0.25 | | |
| | | Umbo | 1.07 | 0.24 | 4.39 | <0.001** |
| | | Low Freq. | -1.07 | 0.19 | -5.57 | <0.001** |
| | | Umbo*Low Freq. | -0.60 | 0.30 | -1.97 | 0.050 |

Sample sizes (n) are presented as the number of temporal bones(velocity measurements). Interaction terms denoted with an asterisk (*). Magnitude models' response variables were log-transformed. Phase models' response variables were transformed by taking the log of the phase plus four. The reference category varied by model and is listed as the first predictor for each model; *t* and *p*-values are not provided for the reference category. Velocity data were extracted from Fig 4 and compiled in S2 Data. See S1 Code for analysis details.

** Denotes a significant difference.

[a] TB19 and TB20 have umbo and stapes measurements, while TB18 has only stapes measurements.

from an arbitrary direction and then using the orthogonal velocity components to calculate a 1D projection onto the piston direction. The similarity of the results between the two methods indicates that the two techniques are mutually supportive and validates this method for making future measurements in elephant.

For ease of comparison across species, Fig 3E–3H shows the same data as in Fig 3A–3D, but grouped according to $V_U/P_{EC}$ (Fig 3E and 3G) and $V_{ST}/P_{EC}$ (Fig 3F and 3H) as measured in elephant (solid lines) and human (dashed lines). The elephant ME resonance occurs at a lower

frequency (near approximately 300 Hz) than in human (1 kHz). As the resonant frequency is determined by both stiffness and mass, the lower resonant frequency in elephant could be due to lower ME stiffness and/or higher ME mass in elephant as compared to human. Below 1 kHz, the $V_U/P_{EC}$ magnitude (Fig 3E) for human was significantly lower than for elephant (p<0.001) by about 5x (14 dB) and up to about 15x (23 dB) near 20 Hz, and the $V_U/P_{EC}$ phases in humans was significantly higher than elephants (p<0.001; Fig 3F). Possible causes of the higher $V_U/P_{EC}$ magnitude in elephant could be related to lower ME stiffness or greater sound collection by the 7.5x -larger elephant TM.

Above 1 kHz, elephant $V_U/P_{EC}$ magnitude decreased by about -6 dB/octave, while for human, it tended to stay relatively flat but was higher than that of elephant above 2 kHz. However, elephant and human $V_U/P_{EC}$ magnitudes above 1 kHz were not significantly different (p = 0.071). At these higher frequencies, the $V_U/P_{EC}$ phase for elephant showed significantly more accumulation than for human (p<0.001; Fig 3G).

Differences in $V_{ST}/P_{EC}$ magnitude between elephant and human ears were very similar to those in $V_U/P_{EC}$ at low frequencies. Below 1 kHz, the elephant $V_{ST}/P_{EC}$ magnitude was higher than that of human (p<0.001), by up to a factor of 5 (14 dB) below the ME resonance frequency of 300 Hz. In contrast, above about 1 kHz, the $V_{ST}/P_{EC}$ magnitudes in the two groups were remarkably similar (p = 0.497).

The $V_{ST}/P_{EC}$ phase for elephant and human reached a peak of up to +0.25 cycles below 200 Hz, consistent with $V_{ST}/P_{EC}$ being approximately dominated by stiffness at low frequencies in both species. At higher frequencies, the slope of the $V_{ST}/P_{EC}$ phase is steeper in elephant than in human (p<0.001), with differences in phase of as much as 2.4 cycles by 10 kHz (Fig 3H), which indicates a greater ME delay than in human, as discussed below.

## Transmission across the ossicular joints

The velocity transfer function from umbo to stapes head $TF_{US}$ is shown in Fig 5A and 5D for both elephant and human. The magnitude of $TF_{US}$ for elephant below 250 Hz is about 0.3 (-10 dB) and is close to the reciprocal of the lever ratio of 2.8 obtained from our Micro-CT anatomical reconstructions (Table 1). Between 250 Hz and 2 kHz, the magnitude of $TF_{US}$ increases to approximately 0.7.

Above 2 kHz the magnitude of $TF_{US}$ increases with frequency, reaching a value of up to 1 in the 3–6 kHz region, and then decreases again. The phase of $TF_{US}$ is about 0 cycles below about 2 kHz, and at higher frequencies it decreases rapidly to -1 cycle by 10 kHz (Fig 5D).

For human, the magnitude of $TF_{US}$ is approximately constant and is about 0.65 (-3.7 dB) below about 1.5 kHz, which is close to 0.7 –the reciprocal of the anatomical LR of 1.4 (Table 1). At higher frequencies the magnitude decreases to about 0.5 in the 2–8 kHz region and then increases to 0.7 near 10 kHz. The $TF_{US}$ phase for human is nearly 0 cycles up to 1 kHz and then decreases to about -0.4 cycles by 10 kHz. These measurements of $TF_{US}$ in human are comparable to measurements reported in Gan et al. [19].

To explore the effects of transmission across the MIC that includes slippage across the IMJ and possible bending of the ossicles, we separately calculated $V_I/V_U$ ($TF_{UI}$) shown in Fig 5B and 5E. We also calculated velocity transfer function from the incus to the stapes head $V_S/V_I$ ($TF_{IS}$) that characterized transmission across the ISJ shown in Fig 5C and 5F. For these measurements, the 3D umbo velocity was projected to a 1D velocity such that it was perpendicular to the plane of the TM. The 3D stapes velocity was also projected to a 1D velocity but now along the piston direction of the stapes. Incus velocity was measured at the end of the incus long process lateral to the stapes head, and thus the incus velocity was also projected along the stapes piston direction. Below about 3 kHz, the magnitude of $TF_{IS}$ is between 0.8–1 for both

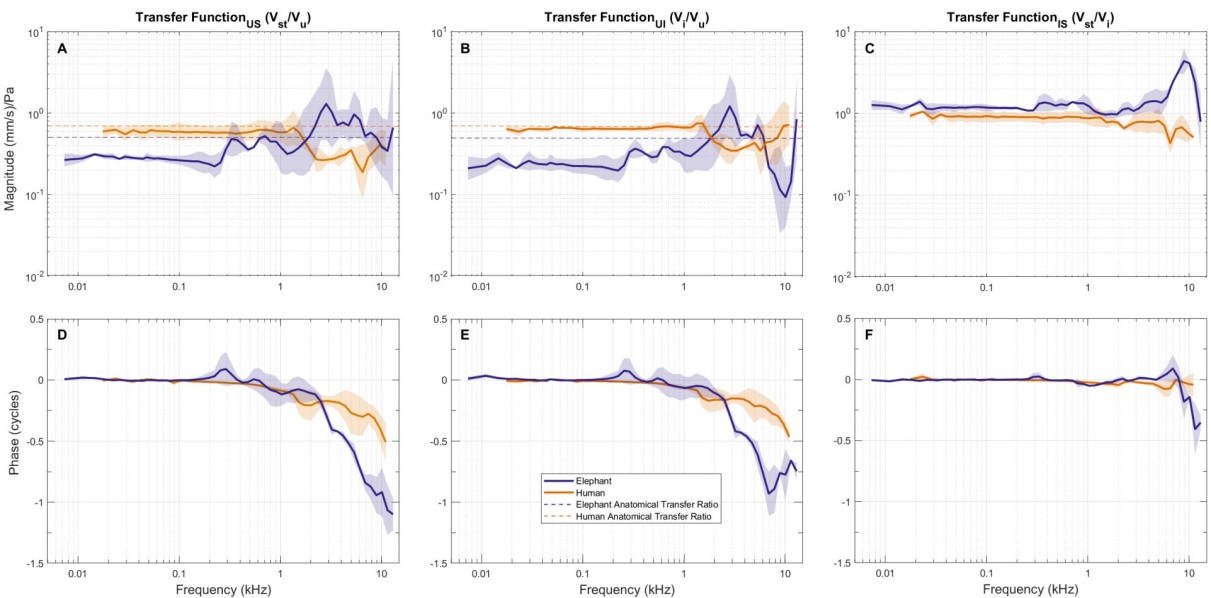

**Fig 5. Vibration transmission across the middle ear bones for both elephant and human.** (A, D) Umbo (M1) to stapes-head transfer function $TF_{US}$. (B, E) Umbo to Incus transfer function $TF_{UI}$ corresponding to transmission across the lever arms and IMJ slippage. (C, F) Incus to stapes transfer function $TF_{IS}$ corresponding slippage across the ISJ. Dashed horizontal lines indicate the reciprocal of the anatomical transfer ratios: human $1/1.4 = 0.7$, and elephant is $1/2 = 0.5$. For I1, M1, two temporal bones for each species with the addition of elephant incus = 2 and human incus = 2). Number of temporal bones: elephant umbo (n = 2), elephant stapes (n = 2), human umbo (n = 2), human stapes (n = 2). Number of total measurements per ossicle per species (elephant umbo = 5, stapes = 7; human umbo = 2, stapes = 4).

elephant and human. Above 5 kHz, this ratio increases by a factor of about 4 for elephant and the phase decreases by about 0.5 cycles, which indicates that there is a resonance between the stapes crus and end of the incus long process. This resonance could be due to the pedicle that connects the incus long process and the incus lenticular plate. Or it could also be due to the ISJ made of a mobile connection between the incus lenticular plate and stapes head.

For human, $TF_{IS}$ magnitude decreased to a factor of about 0.5 for humans, and the phase variation was less than 0.1 cycles throughout the measured frequency range. Velocity transmission across the MIC $TF_{UI}$ (Fig 5B and 5E) appear very similar to overall transmission from the umbo to stapes $TF_{US}$ (Fig 5A and 5D) below about 5 kHz. These results indicate that the behavior of the elephant ossicular chain is primarily determined by the mechanics of the MIC, as indicated by $TF_{UI}$ below about 5 kHz, with a possible contribution due to resonance across the ISJ above 5 kHz.

## Middle-ear group delays

The wideband $V_{ST}/P_{EC}$ group delay (GD) (Table 3) for elephant was 2.5x greater than for human (138 vs 56 μs), and the wideband $V_U/P_{EC}$ GD for elephant was 4.3x greater than for human (61 vs 14 μs). In the high-frequency region, the $V_{ST}/P_{EC}$ GD for elephant was 2.3x that of human (136 vs 60 μs), while the $V_U/P_{EC}$ GD for elephant was 6.2x higher (59 vs 9.5 μs). There was little difference in the GDs in the near-resonance region. The GD for $V_{ST}/P_{EC}$ was higher than that for $V_U/P_{EC}$ by a factor of about 2.3 for elephants for both HF and WB and for humans 6.3 and 4, respectively. Transmission delays for elephant were nearly the same for both HF and WB (77.5 and 77.2) and for human (50.5 and 42, respectively), the elephant being 1.5x higher for HF and 1.8x higher for WB.

**Table 3. Group delays in microseconds, calculated for elephants and humans.**

| | | Elephant | | | Human | |
|---|---|---|---|---|---|---|
| | Specimen | HF (1200–11000) | WB (600–11000) | Specimen | HF (4000–11000) | WB (2000–11000) |
| Umbo | ETB2 | 5 | 3 | TB18 | - | - |
| | ETB3 | -2 | 2 | TB19 | 10 | 14 |
| | ETB6 | 170 | 161 | TB20 | 9 | 14 |
| | ETB7 | 62 | 77 | | | |
| | *Mean* | *58.8* | *60.8* | *Mean* | *9.5* | *14.0* |
| | SD | 68.9 | 65.4 | SD | 0.5 | 0 |
| Stapes | ETB2 | 121 | 111 | TB18 | 88 | 73 |
| | ETB3 | 104 | 111 | TB19 | 50 | 41 |
| | ETB6 | - | - | TB20 | 42 | 54 |
| | ETB7 | 184 | 192 | | | |
| | *Mean* | *136.3* | *138.0* | *Mean* | *60.0* | *56.0* |
| | SD | 34.4 | 38.2 | SD | 20.1 | 13.1 |

All frequencies are displayed in Hz. HF = high frequency, WB = wideband, TB = human temporal bone, ETB = elephant temporal bone, SD = standard deviation. Group delay windows were chosen so they are all above middle ear resonance frequencies (as described in Methods). Group delays were calculated using phase data from Fig 4. To accommodate for a 5 mm ear canal length in humans, from probe tube to umbo, we subtracted 15 μs from each human measurement, and since there was a large age difference between elephants, hence differences in ear canal length, we subtracted 119 μs for ETB2; 154 μs for ETB3; 55μs for ETB6 and 131 μs for ETB7 individually prior to calculating the means.

## Discussion

In this study of the largest terrestrial ear, we quantify the sound-transmission properties of the elephant ME with acoustic stimulation and compare those results to human. Most of the results are consistent with current understanding of ME mechanics, but we also had some unexpected results.

### Differences in ME response relative to anatomy

Both $V_U/P_{EC}$ and $V_{ST}/P_{EC}$ in both elephant and human showed a resonance, near 300 Hz in elephant and ~1000 Hz in human. Below resonance, $V_U/P_{EC}$ and $V_{ST}/P_{EC}$ increased with frequency, with a slope of about +6 dB/octave. At frequencies below resonance in elephant, $V_U/P_{EC}$ and $V_{ST}/P_{EC}$ magnitudes in elephant were 5–10x (14–20 dB) greater than in human (Fig 3E and 3F).

Possible reasons for a greater ME response at lower frequencies include a higher force at the umbo for the same ear-canal sound pressure. At low frequencies, where $P_{EC}$ is constant over the TM and the TM can be assumed to move as a unit [20], umbo force $F_U = Atm \times *P_{EC}$, where Atm is the TM area. The elephant eardrum area is about 7.5x that of human (Table 1), which means the force at the umbo for elephant should be 7.5x higher than in human. This 7.5x higher force is consistent with the higher $V_U$ in elephant relative to human. Impedance at the umbo could also be lower, such that the same $P_{EC}$ would cause more motion. The umbo impedance in elephant is unknown, however.

The ME resonance observed in $V_U/P_{EC}$ in both elephant and human allows us to estimate the effective ME mass. To a first-order approximation, the ME-resonance frequency $f_{me} = \sqrt{\frac{K_{me}}{m_{me}}}$, where $K_{me}$ is the ME stiffness and $m_{me}$ is the effective ME mass, including the TM and ossicles. Note that the effective ossicular mass for resonance calculations may be smaller than the actual physical mass of the ossicles due to the geometry of the middle ear.

We assume that the elephant ME stiffness is similar to that of human, because the difference in $V_U/P_{EC}$ elephant and human can be explained by the difference in $F_U$ (from above). In this case, the lower resonant frequency in elephant is due to a higher effective ME mass in elephant and suggests that the $m_{me}$ in elephant is about 10x that of human, which is consistent with the masses reported in Table 1, the elephant being ~9x that of human. Similarly, the eardrum mass average among elephant species is about 14x greater than human, largely because the elephant TM surface area was about 7 times bigger and twice as thick. Thus, the lower elephant ME resonance frequency than human is likely due to the higher combined mass of the TM and ossicles.

At higher frequencies, $V_U/P_{EC}$ magnitude is very different between elephant and human. Above the human ME resonance at about 1 kHz, $V_U/P_{EC}$ remains roughly constant (within a range of 0.1 to 0.2 (mm/s)/Pa), while in the elephant $V_U/P_{EC}$ decreases with a slope of approximately –6 dB/oct. This indicates that elephant umbo velocity is mass-limited at frequencies above 1.5kHz.

$V_{ST}/P_{EC}$ shows a lower magnitude and similar frequency response to $V_U/P_{EC}$ below the ME resonance frequency for both elephant and human (Fig 3A and 3B). However, for frequencies around 5 or 6 kHz, $V_{ST}/P_{EC}$ in elephant is larger than $V_U/P_{EC}$ (Fig 3A), while in human $V_{ST}/P_{EC} < V_U/P_{EC}$ at all frequencies (Fig 3B). This is surprising given the expectation that species with more massive ossicles would have reduced sensitivity at higher frequencies.

In contrast, at frequencies above about 1 kHz, the magnitudes of $V_{ST}/P_{EC}$ for both elephant and human are surprisingly similar (Fig 3F). This was unexpected as well, given that the estimated effective mass was 10x larger in elephant than in human, which would result in 10x greater ME impedance, and thus reduce transmission at the higher frequencies by about the same amount for a given umbo force. The similarity in stapes velocities may be facilitated by the 7.5x higher force at the umbo due to the 7.5x larger TM area of the elephant.

## Middle-ear group delay

One of the important findings of this study is the large phase accumulation and corresponding GD of the elephant $V_U/P_{EC}$ and $V_{ST}/P_{EC}$ responses (Table 3). We discuss the results in the framework proposed by Puria and Allen [10], who decomposed the ME delay into TM and ossicular-chain delays, where TM delay is shown by $V_U/P_{EC}$ delay and ossicular-chain delay is shown by the additional delay in $V_{ST}/P_{EC}$ over $V_U/P_{EC}$. They proposed that TM delay shows how mass inertia of the TM, coupled with its stiffness from radial collagen fibers, forms a transmission line to minimize high-frequency loss due to TM mass inertia. Similarly, they proposed that ossicular delay indicates that the mass of the ossicles, coupled with ossicular and joint stiffness, also form a transmission line to minimize high-frequency mass-limiting of $V_{ST}/P_{EC}$ due to the ossicles. Here, we similarly analyze our GD results calculated at the umbo and stapes.

The HF and WB delays in elephant $V_U/P_{EC}$ were very similar at about 60 µs and about 5x higher than delays in human $V_U/P_{EC}$ (10–14 µs). The radius of the elephant TM is about 3x larger than that of human (13.6 vs. 5.0 mm). This result suggests that the wave speed (computed by dividing TM radius by delay [10]) in elephant (226 m/s) is about half that in human (425 m/s). This may be consistent with the thicker and more massive TM in elephant vs. human. Our estimated TM delay of 60 µs in elephant is about 1.2x (O'Connor and Puria [21]; 50 µs) to 5x (Table 3, 12 µs) that of the human TM.

The ossicular-chain delay is computed by subtracting $V_U/P_{EC}$ from $V_{ST}/P_{EC}$ delay. The large $V_{ST}/P_{EC}$ phase accumulation seen in Fig 3H is revealed in a greater $V_{ST}/P_{EC}$ GD (138 µsec) in elephant than human (56 µsec). The WB ossicular-chain delay is 1.8x larger in elephant (77 µsec) than human (42 µsec), which is consistent with the higher mass of the elephant ossicles but also perhaps indicative of greater looseness in the ossicular joints.

### Relative motion across the ossicular joints

As discussed above, the umbo-to-stapes velocity transfer function $TF_{US}$ in elephant and human was close to the reciprocal of the anatomical LR at low frequencies, and the measured phase was close to zero cycles (Fig 5A and 5D), as expected from the anatomy. At higher frequencies, $TF_{US}$ in human decreased, as has been seen previously in other species; but $TF_{US}$ in elephant increased, which was unexpected.

Magnitude attenuation in $TF_{US}$ at higher frequencies has been reported in other animals such as cat for frequencies above 10 kHz [10, 22], and gerbil and chinchilla for frequencies above about 30 kHz [23, 24]. These attenuations in $TF_{US}$ are generally attributed to flexibility in the IMJ, with the additional flexibility possibly providing some degree of protection against high-amplitude sounds (reviewed in [25]). It has been further shown that ME-joint flexibility reduces the peak amplitude of impulsive sounds by spreading their energy out over time, which may also serve to protect sensitive cochlear structures [8].

In elephant, there is an abrupt rise in the magnitude at ME resonance near 300 Hz by a factor of 2, with a corresponding increase in phase starting at about 150 Hz. The $TF_{US}$ magnitude then stays roughly in the vicinity of 0.4–0.5 from 400 Hz to 1.3 kHz. Surprisingly, the $TF_{US}$ magnitude increases again between 2 and 8 kHz, reaching a peak of about 1 (considerably higher in some specimens), and decreases again at higher frequencies. Above 2 kHz the phase decreases rapidly with frequency, reaching -0.5 cycles at about 5 kHz and continuing to decrease to -0.9 cycles by 10 kHz. The increase in $TF_{US}$ magnitude and decrease in $TF_{US}$ phase may be related, as discussed below.

### Possible reasons for increased transmission along the elephant malleus and incus

The increase in elephant $TF_{US}$ magnitude above about 250 Hz looks very different from data in human, cat, and gerbil, where it generally decreases with increasing frequency. A possible explanation comes from Rosowski et al. [26], who measured the surface motion of chinchilla ossicles from 0.5 to 18 kHz using Optical Coherence Tomography (OCT) Vibrometry. They calculated $TF_{US}$ similarly (but in terms of a displacement ratio "$X_{Footplate}/X_{Umbo}$") and also reported the motion of various points along the length of the malleus relative to umbo motion. Their measurements were surprisingly similar to our measurements in elephant but scaled down in frequency by about a factor of 10. They observed an increase in the incus tip displacement relative to the umbo (comparable to $TF_{UI}$) above about 5 kHz, whereas for elephant this occurs near about 200 Hz for $TF_{UI}$. In addition, the chinchilla $TF_{US}$ starts to increase near about 10 kHz, reaching a peak near 14 kHz, with an increase in phase from 0 cycles at low frequencies to about -0.5 cycles near 18 kHz. In elephant, the $TF_{US}$ magnitude started to increase near 1 kHz and had a broad peak in the 2–8 kHz region, with the phase at -0.5 cycles at about 6 kHz. Near 10 kHz, there is a resonance in $TF_{IS}$ (Fig 5C and 5F) due to the elephant pedicle that joins the incus to the ISJ or due to the ISJ itself.

Rosowski et al. [26] identified three different modes of vibration in the chinchilla ossicles: 1) rotation around the classical anterior–posterior axis below 8 kHz, 2) lateral-to-medial translation from 2 to 10 kHz and beyond, and 3) a new rotational mode around a superior–inferior axis that parallels the manubrium and appears to be dominant between 10 and 16 kHz. Given the close parallels between $TF_{US}$ in chinchilla and elephant, the same modes in chinchilla might also be evoked in the elephant ossicles, after scaling for frequency.

The ossicles of chinchilla and elephant exhibit many morphometric differences. The most obvious is that the elephant ossicular mass is about 100x heavier than that of chinchilla, for which the mass is about 5 mg. The square root of the mass ratio is about 9.2, close to the

frequency scale factor of around 10 between the two species. The IMJ in the chinchilla becomes fused in adults, and thus the malleus and incus function as a single sausage-shaped unit with malleus and incus lever arms protruding perpendicular to the long axis of the sausage. This shape was hypothesized to give rise to a novel second mode of vibration along an inferior–superior axis that boosts $TF_{US}$ by up to a factor of 2–3 above 10 kHz, resulting in a -0.5 cycle out of phase motion between the umbo and stapes near the peak of about 14 kHz. Our observation of a similar increase in elephant ossicular transmission starting at about 1 kHz, along with a phase lag of -0.5 cycles near 6 kHz (Fig 5), suggests there may be a similar mode of vibration in the elephant ossicles. This would also suggest that the IMJ is fused in elephants. However, manual palpitations and histology data (unpublished) indicate that the elephant IMJ is quite mobile, suggesting that the underlying modes of vibration between the two species might be very different.

Our observation of a possible boost in elephant $TF_{IS}$ due to an incus pedicle resonance and/or ISJ has not been observed in human, cat, and gerbil [27–29]. Robles et al. [30] and Ruggero et al. [31] reported a boost in transmission across $TF_{IS}$ of chinchilla. However, they later suggested that such a boost was dependent on a measurement angle that was off-axis from the piston direction. By calculating a stapes piston direction for our 3D motion data, we remove this uncertainty from the $TF_{IS}$ calculation showing a boost near 9 kHz. Finite element models [32, 33] indicate that the pedicle might play a role in reducing non-piston like motions from entering the cochlea. Our measurements of $TF_{IS}$ might be the first to show a boost in transmission across the incus pedicle and ISJ. One caveat is that errors due to projections could also result in a boost in $TF_{IS}$, and this needs further investigation.

## TM and ossicular-chain morphology and potential protective aspects

The elephant malleus cross section appears to be shaped like an oval with a high major to minor axis ratio, otherwise termed oval eccentricity (see S1B Fig and Fig 2B and 2C). In addition, the malleus divides the elephant TM symmetrically (S1B Fig), similar to a chinchilla, gerbil and guinea pig [16], a configuration referred to as an I-beam, often used in building construction because they are very resistant to bending. The human and cat malleus, however, have a more-or-less circular cross section with the TM divided asymmetrically by the malleus, which can better withstand twisting modes [16]. The fact that the elephant, with the greatest known eardrum area with likely the largest forces at the umbo, possesses a malleus shaped similar to an I-beam rather than circular, is consistent with the idea that the I-beam may serve a protective purpose.

**Unanswered questions remain.** Given the mass of elephant ossicles, the mechanisms enabling higher-frequency hearing, other than a large eardrum area, are still not clear. The possibility of additional modes of ossicular motion at higher frequencies could explain this result but requires additional research.

## Evolutionary benefits of elephant low-frequency hearing

Elephants communicate using low frequency vocalizations that propagate long distances, making it possible to maintain contact with extended family members [34, 35], share information about the environment and potential danger [36–38], as well as advertising reproductive states [35, 39, 40]. The ability to hear these low frequency sounds enables the elephant to maintain complex social relationships across many different landscapes.

The advantages of being able to communicate over long distances would have benefited elephants over evolutionary time. For the genus *Loxodonta*, as the climate became drier in sub-Saharan Africa and the forest landscape shifted to a more open savannah, the ability to detect

low-frequency sounds would have allowed individuals to remain in contact with family members while still being able to feed across longer distances. Since higher frequencies attenuate in dense forest habitats, where the genus *Elephas* evolved, low frequency vocalizations may have been selected for to facilitate communication in both the near and far field. The results of this study support that there likely was pressure for the middle ear to evolve along with low-frequency vocalizations to facilitate low-frequency communication. Over evolutionary time, there was also pressure for the elephant cochlear structures and central mechanisms to evolve to enable low frequency hearing. There is still much we don't know about elephant communication and evolutionary pressures on ancestral species, limiting our ability to make definitive statements about evolution.

## Conclusions

This study quantifies elephant auditory anatomy and function obtained using modern measurement techniques. New details of ME anatomy in both African and Asian elephant are described, and ME function is quantified for the first time. Many aspects of elephant ME function are comparable to those observed in human but at lower frequencies, consistent with the larger size and mass of elephant ME components. At the same time, it appears that the larger size of the elephant TM surface area facilitates greater low-frequency sound collection and perhaps better ME function at higher frequencies as well, partly overcoming the effects of greater ME effective mass.

We also observed several features in elephant ME function not described previously in any species. The significantly larger GD in elephant ears in comparison to human ears suggests that delay may play a role in improved high-frequency hearing, especially when more mass is involved as previously hypothesized [10]. In addition, the incus pedicle and ISJ may boost the stapes through resonance near 9 kHz. Further studies on GD, as well as studies on a broader range of mammalian species, may offer additional insights into the general principles governing the evolution and adaptation of the mammalian auditory system.

## Supporting information

**S1 Fig. Conversion of velocities Vx,y,z measured in the LDV coordinate frame to the anatomical 1D piston direction.** (A) Stapes piston direction (green arrow) calculate from the 3D LDV reference measurement and angle measurements. The measurement was made at the stapes head. (B) The measured 3D umbo velocity was projected to an umbo 1D velocity (red arrow) along a vector normal to the plane of the TM (the black line lies on this plane). The stapes velocity along its piston direction is shown by the green arrow. µCT reconstructions are for ETB2.
(TIF)

**S2 Fig. Demonstration of the equivalence of rotations applied to transform the 3D LDV (extrinsic) reference frame to the stapes (intrinsic) reference frame.** All panels show the extrinsic (*x*-*y*-*z*) reference frame in black, superposed over a photograph of the stapes and surroundings taken by the camera through the dichroic mirror on the 3D LDV and rotated slightly to show the *z* axis more clearly. The origin is at the center of the stapes footplate. (A-D) Rotations in TB20 in the order they were measured; (E-H) Rotations in the order used by the Spatial Math Toolbox. (A) Extrinsic reference frame; (B) rotation of +20˚ about the *z* axis (blue arrow) to align the *y* axis (green) to the projection of the stapes piston direction in the *x*-*y* plane–the rotated *x* and *y* axes are shown in red and green respectively; (C) additional rotation of +40˚ about the (new) *x* axis (red arrow) = elevation from the *x*-*y* plane–the rotated *z*

axis is shown in blue; (D) additional rotation of –150˚ about the (new) *y* axis (green arrow) to arrive at the *Maj-pist-min* intrinsic reference frame (red-green-blue). (E) The extrinsic coordinate system as in (A); (F) rotation of –139˚ about the *z* axis (blue arrow); (G) additional rotation of –22˚ about the (new) *y* axis (green arrow); (H) additional rotation of +135˚ about the (new) *x* axis (red arrow). Panels (D) and (H) are identical, which shows that the two rotation sequences are equivalent.
(TIF)

**S1 Table. Projection rotation angles (in degrees) for each temporal bone, as measured (Azimuth, elevation, rotation) and as recomputed for implementation in Z-Y-X order.** Two orientations were used for ETB2: A with the tympanum proper intact (more posterior), and B with the tympanum proper removed (more anterior). Separate projections were applied to stapes footplate and incus velocities (SFP) and umbo velocities (UMB), as the stapes and umbo piston velocities were in slightly different directions. ETB7 Rz is the same as Azimuth.
(DOCX)

**S1 Text. Further details on elephant specimen preparation methods.**
(DOCX)

**S2 Text. Measurements of 3D ossicle velocities.**
(DOCX)

**S3 Text. How piston directions were calculated.**
(DOCX)

**S4 Text. Computation of velocities in anatomically relevant piston directions.**
(DOCX)

**S1 Data. Raw velocity data for all specimens.**
(ZIP)

**S2 Data. Velocity data extracted from Fig 4 after all methodological processing (see Methods, S3 and S4 Texts) used for statistical analysis conducted using S1 Code.**
(CSV)

**S1 Code. R code used for statistical analysis of S2 Data.**
(TXT)

## Acknowledgments

We thank Jeffrey Tao Cheng and John J. Rosowski for edits and comments. Elephant postmortem specimens were donated by the following zoos: Pittsburgh Zoo & Aquarium, National Zoo, St. Louis Zoo, Santa Barbara Zoo, Oakland Zoo, San Antonio Zoo, Dallas Zoo, and the White Oak Sanctuary. We thank the reviewers for their helpful feedback.

## Author Contributions

**Conceptualization:** Caitlin E. O'Connell-Rodwell, Xiying Guan, Sunil Puria.

**Data curation:** Jodie L. Berezin, Yihan Hu, Kevin N. O'Connor.

**Formal analysis:** Jodie L. Berezin, Yihan Hu.

**Funding acquisition:** Caitlin E. O'Connell-Rodwell, Sunil Puria.

**Investigation:** Caitlin E. O'Connell-Rodwell, Jodie L. Berezin, Anbuselvan Dharmarajan, Michael E. Ravicz, Xiying Guan, Sunil Puria.

**Methodology:** Caitlin E. O'Connell-Rodwell, Anbuselvan Dharmarajan, Michael E. Ravicz, Xiying Guan, Kevin N. O'Connor, Sunil Puria.

**Project administration:** Caitlin E. O'Connell-Rodwell.

**Resources:** Caitlin E. O'Connell-Rodwell, Anbuselvan Dharmarajan, Michael E. Ravicz, Sunil Puria.

**Software:** Kevin N. O'Connor.

**Supervision:** Caitlin E. O'Connell-Rodwell, Sunil Puria.

**Validation:** Caitlin E. O'Connell-Rodwell, Anbuselvan Dharmarajan, Michael E. Ravicz, Xiying Guan, Sunil Puria.

**Visualization:** Jodie L. Berezin, Michael E. Ravicz, Yihan Hu.

**Writing – original draft:** Caitlin E. O'Connell-Rodwell, Jodie L. Berezin, Michael E. Ravicz, Sunil Puria.

**Writing – review & editing:** Caitlin E. O'Connell-Rodwell, Jodie L. Berezin, Anbuselvan Dharmarajan, Michael E. Ravicz, Kevin N. O'Connor, Sunil Puria.

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
