## [Decision Letter · Decision Letter 0]

19 Dec 2023

PONE-D-23-31155The impact of size on middle-ear sound transmission in elephants, the largest terrestrial mammalPLOS ONE

Dear Dr. O'Connell-Rodwell,

Thank you for submitting your manuscript to PLOS ONE. After careful consideration, we feel that it has merit but does not fully meet PLOS ONE’s publication criteria as it currently stands. Therefore, we invite you to submit a revised version of the manuscript that comprehensively addresses the points of both reviewers.

We look forward to receiving your revised manuscript.

Kind regards,

Andre van Wijnen

Academic Editor

PLOS ONE

“We thank Jeffrey Tao Cheng and John J. Rosowski for edits and comments. Elephant postmortem specimens were donated by the following zoos: Pittsburgh Zoo & Aquarium, National Zoo, St. Louis Zoo, Santa Barbara Zoo, Oakland Zoo, San Antonio Zoo, Dallas Zoo, and the White Oak Sanctuary. Work supported by K01 DC017812 from the NIDCD of NIH (to COR) and The Amelia Peabody Fund (to SP).”

 “Work supported by K01 DC017812 from the NIDCD of the National Institute of Health (to CEO-R) and The Amelia Peabody Fund (to SP).

6. We are unable to open your Supporting Information file [S1_Data.zip]. Please kindly revise as necessary and re-upload.

Reviewers' comments:

Reviewer's Responses to Questions

**Comments to the Author**

1. Is the manuscript technically sound, and do the data support the conclusions?

Reviewer #1: Yes

Reviewer #2: Yes

2. Has the statistical analysis been performed appropriately and rigorously? 

Reviewer #1: Yes

Reviewer #2: Yes

3. Have the authors made all data underlying the findings in their manuscript fully available?

Reviewer #1: Yes

Reviewer #2: Yes

4. Is the manuscript presented in an intelligible fashion and written in standard English?

Reviewer #1: Yes

Reviewer #2: Yes

5. Review Comments to the Author

Reviewer #1: This manuscript presents an interesting study to compare middle ear transfer function (i.e. velocities of the umbo and the ossicles in response to ear canal sound pressure) between elephant and human. The velocity was measured with modern Laser Doppler Vibrometry (LDV) technique across a broad frequency range between 7 and 13,000 Hz. The results suggest elephant has a lower middle ear resonance frequency of about 300 Hz than human around 1000 Hz, which may facilitate elephant long distance communication. Surprisingly, despite larger size and heavier mass of the elephant middle ear, sound transmission through elephant ear at high frequencies is boosted to the similar level of human ear (see Fig.2E and 2F, where despite umbo motions in Elephant above 2 kHz are lower than those in Human, stapes motions in Elephant are close to those in Human). This is surprising as it counteracts the mass effect at high frequencies. The authors attempted to explain this surprising finding through several unique morphological features of the elephant middle ear (such as “an incus pedicle resonance and/or ISJ”) as well as potential changes of its vibration modes (like those observed in chinchillas in a recent study) at high frequencies. Another surprising finding is the significantly larger group delay (phase accumulation) in elephant ears in comparison to human ears, “which may play a role in improved high-frequency hearing, especially when more mass is involved”, as being speculated in the manuscript by the authors.

Line 77: “whereas it was reported as 9.6x larger (454 vs. 68.3 mm2)”. It should be “6.6x” larger.

In Table 1, the TM surface areas between African and Asian Elephants are significantly different (606.3 vs. 403.6), suggesting species differences. Indeed the TM surface area of Asian Elephant is not that much different from Nummela et al.

How was the thickness of the TM quantified? Given the thickness of the human TM quantified in this study (223 um) is more than twice of the reported human TM thickness (80~100 um), please give more details and justify the results.

Is the density of elephant middle ear same as the density of human ear when calculating the mass?

In Table 3, some mean values do not seem to match with individual values listed there. For instance, in Human HF column, how could 25 and 24 yield mean of 9.5? Please double check all your calculations of means and SDs and revise text accordingly.

Given only a limited number of elephant ears were measured in this study, would it make more sense to look at individual ear rather than looking at the mean and SD? For example, in Table 3, in Elephant HF column, the umbo group delay from four elephant ears are 5, -2, 170, 62, which makes no sense to average them. Plots like those shown in S2_Fig are more meaningful.

Reviewer #2: The authors carefully evaluated and described the middle ear function of elephants and compared it to 4 human temporal bones. Although the number of human temporal bones is quite small, I only have minor comments:

-Line 384: “Fig. 2H” instead of “Fig. 3H”

- Can you transfer your findings to the real life of elephants and speculate on the evolutionary benefits of their hearing?

- Are the findings of the two species (African and Asian elephant) comparable? Are the differences species or more age related?

6. PLOS authors have the option to publish the peer review history of their article (what does this mean?). If published, this will include your full peer review and any attached files.

Reviewer #1: No

Reviewer #2: No

---

## [Author Response · Author response to Decision Letter 0]

8 Jan 2024

Responses to reviewer and editor comments are in the "Response to Reviewers" document (labeled as a rebuttal letter).

---

## [Decision Letter · Decision Letter 1]

18 Jan 2024

PONE-D-23-31155R1The impact of size on middle-ear sound transmission in elephants, the largest terrestrial mammalPLOS ONE

Dear Dr. O'Connell-Rodwell,

Thank you for submitting your manuscript to PLOS ONE. While the manuscript has been improved, one of the reviewers has requested additional minor modifications. Your paper is essentially provisionally accepted pending these final minor changes. 

We look forward to receiving your revised manuscript.

Kind regards,

Andre van Wijnen

Academic Editor

PLOS ONE

Journal Requirements:

Reviewers' comments:

Reviewer's Responses to Questions

**Comments to the Author**

1. If the authors have adequately addressed your comments raised in a previous round of review and you feel that this manuscript is now acceptable for publication, you may indicate that here to bypass the “Comments to the Author” section, enter your conflict of interest statement in the “Confidential to Editor” section, and submit your "Accept" recommendation.

Reviewer #1: (No Response)

Reviewer #2: All comments have been addressed

2. Is the manuscript technically sound, and do the data support the conclusions?

Reviewer #1: Yes

Reviewer #2: Yes

3. Has the statistical analysis been performed appropriately and rigorously? 

Reviewer #1: Yes

Reviewer #2: Yes

4. Have the authors made all data underlying the findings in their manuscript fully available?

Reviewer #1: Yes

Reviewer #2: Yes

5. Is the manuscript presented in an intelligible fashion and written in standard English?

Reviewer #1: Yes

Reviewer #2: Yes

6. Review Comments to the Author

Reviewer #1: I appreciate the efforts the authors have made into their revisions. I only have a few minor suggestions.

Line 216-217: “Nummela [2] reported human, seal, and elephant ossicle densities between 2.0 and 2.3.” Please add the unit of density after the numbers.

Table 3: Thanks for clarifying that you subtracted 15us to calculate human means. Looks like you also subtracted different times (varying from specimen to specimen) when calculating elephant means. However, are the values listed in Table 3 in the elephant columns from “after subtraction”? While the values listed in Table 3 in the human columns are from “before subtraction”? This is confusing, and I suggest you also list the values for the humans after subtraction.

Reviewer #2: The authors responded to all the reviewer's comments. I do not have any further comment and I would like to congratulate the authors for their excellent work.

7. PLOS authors have the option to publish the peer review history of their article (what does this mean?). If published, this will include your full peer review and any attached files.

Reviewer #1: No

Reviewer #2: No

---

## [Author Response · Author response to Decision Letter 1]

19 Jan 2024

Please see the Rebuttal Letter for responses to the editor and reviewers. Thank you.

---

## [Editor Report · Decision Letter 2]

26 Jan 2024

The impact of size on middle-ear sound transmission in elephants, the largest terrestrial mammal

PONE-D-23-31155R2

Dear Dr. O'Connell-Rodwell,

We’re pleased to inform you that your manuscript has been judged scientifically suitable for publication and will be formally accepted for publication once it meets all outstanding technical requirements.

Kind regards,

Andre van Wijnen

Academic Editor

PLOS ONE

Additional Editor Comments (optional):

Editorial Comments:

The authors have adequately addressed the final minor comments.
---

## [Editor Report · Acceptance letter]

19 Mar 2024

PONE-D-23-31155R2 

PLOS ONE

Dear Dr. O'Connell-Rodwell, 

I'm pleased to inform you that your manuscript has been deemed suitable for publication in PLOS ONE. Congratulations! Your manuscript is now being handed over to our production team.

Kind regards, 

on behalf of

Dr. Andre van Wijnen 

Academic Editor

PLOS ONE